# Dynamic neuromuscular remodeling precedes motor-unit loss in a mouse model of ALS

Éric Martineau[1,2], Adriana Di Polo[1,3], Christine Vande Velde[1,3], Richard Robitaille[1,2]*

[1]Département de neurosciences, Université de Montréal, Québec, Canada; [2]Groupe de recherche sur le système nerveux central, Université de Montréal, Québec, Canada; [3]Centre de recherche du Centre Hospitalier de l'Université de Montréal, Québec, Canada

**Abstract** Despite being an early event in ALS, it remains unclear whether the denervation of neuromuscular junctions (NMJ) is simply the first manifestation of a globally degenerating motor neuron. Using in vivo imaging of single axons and their NMJs over a three-month period, we identify that single motor-units are dismantled asynchronously in $SOD1^{G37R}$ mice. We reveal that weeks prior to complete axonal degeneration, the dismantling of axonal branches is accompanied by contemporaneous new axonal sprouting resulting in synapse formation onto nearby NMJs. Denervation events tend to propagate from the first lost NMJ, consistent with a contribution of neuromuscular factors extrinsic to motor neurons, with distal branches being more susceptible. These results show that NMJ denervation in ALS is a complex and dynamic process of continuous denervation and new innervation rather than a manifestation of sudden global motor neuron degeneration.

DOI: https://doi.org/10.7554/eLife.41973.001

*For correspondence:
richard.robitaille@umontreal.ca

**Competing interests:** The authors declare that no competing interests exist.

## Introduction

Amyotrophic lateral sclerosis (ALS) is a fatal neurodegenerative disease characterized by upper and lower motor neuron (MN) loss. Loss of neuromuscular junctions (NMJs) is a key pathological event in ALS patients (*Fischer et al., 2004*; *Killian et al., 1994*) and in animal models of the disease (*Clark et al., 2016*; *Fischer et al., 2004*; *Frey et al., 2000*; *Pun et al., 2006*; *Tallon et al., 2016*). Denervation of NMJs is observed prior to symptom onset (*Clark et al., 2016*; *Fischer et al., 2004*; *Pun et al., 2006*; *Tallon et al., 2016*; *Vinsant et al., 2013*) and before any significant MN axonal or cell body loss (*Fischer et al., 2004*; *Vinsant et al., 2013*). However, whether early NMJ denervation arises from local branch-specific degeneration ('dying-back hypothesis') or is merely the manifestation of a diseased and globally degenerating motor neuron remains ill-defined.

There is considerable evidence which suggests that a subset of degenerating motor-units (MU, a motor neuron and the muscle fibers it innervates) swiftly lose their NMJs while a distinct, yet unaffected, population compensates via axonal sprouting (*Pun et al., 2006*; *Schaefer et al., 2005*), somewhat akin to a nerve injury. Indeed, NMJs innervated by fast-fatigable MNs are lost early, followed by fast-fatigue resistant and slow MNs (*Frey et al., 2000*; *Pun et al., 2006*). This is further supported by electromyographic (EMG) recordings showing that the number of MUs declines in parallel with muscle strength (*Azzouz et al., 1997*; *Dantes and McComas, 1991*; *Hegedus et al., 2007*; *Hegedus et al., 2008*; *Kennel et al., 1996*; *McComas et al., 1971*) and that surviving MUs increase in size in animal models and in patients (*Dantes and McComas, 1991*; *Dengler et al., 1990*; *McComas et al., 1971*; *Schmied et al., 1999*). Taken together, these results strongly suggest

**eLife digest** Amyotrophic lateral sclerosis (ALS), also known as Lou Gehrig's disease, is a fatal neurodegenerative disorder. It occurs when the neurons that control muscles – the motoneurons – disconnect from their target muscles and die. This causes the muscles to weaken and waste away. More and more muscles become affected over time until eventually the muscles that control breathing also become paralyzed. Most patients die within two to five years of diagnosis.

Motoneurons consist of a cell body plus a cable-like structure called the axon. The cell body of each motoneuron sits within the spinal cord, and the axon extends out of the spinal cord to the motoneuron's target muscle. Within the muscle the axon divides into branches, each of which connects with multiple muscle fibers. The breakdown of these connections, known as neuromuscular junctions, is one of the first signs of ALS.

Does a motoneuron lose all of its connections with muscle fibers at once, or do the connections break down a few at a time? This distinction is important as it will help to identify the events that lead to muscle paralysis in ALS. To find out, Martineau et al. studied mice that had two genetic mutations: one that causes ALS and another that produces fluorescent molecules in some motoneurons. This allowed the branches of the motoneurons to be tracked over time with a fluorescence microscope.

Martineau et al. found that individual neurons lose their connections to muscle fibers gradually. Moreover, motoneurons grow new branches and form new connections even while losing their old ones. This dual process of pruning and budding lasts for several weeks, until eventually the motoneuron dies.

Developing drugs to stabilize neuromuscular junctions during the period when motoneurons gradually disconnect from muscles could be a promising avenue to explore to improve the quality of life of ALS patients. One advantage of this treatment strategy is that neuromuscular junctions in muscles are easier to access than motoneurons inside the spinal cord. To identify potential drugs, future studies will need to focus on the proteins and signals that cause the neuromuscular junctions to break down.

DOI: https://doi.org/10.7554/eLife.41973.002

that the loss of NMJs within a single MU is a dichotomous and mostly synchronous event and that local synaptic events have a limited impact on the denervation process. Under this monotonic paradigm, degenerating MUs are irrevocably caught in a degenerative cascade while surviving MUs attempt to compensate, consequently undermining the relevance of the NMJ as a disease-modifying therapeutic target.

However, the rescue of neuronal loss in SOD1 mice does not necessarily improve NMJ denervation (*Gould et al., 2006*; *Parone et al., 2013*; *Suzuki et al., 2007*) suggesting that local branch-specific events may play a key role in NMJ denervation. In ALS, local alterations of synaptic function and molecular signaling take place very early at the NMJ (*Arbour et al., 2015*; *De Winter et al., 2006*; *Jokic et al., 2006*; *Rocha et al., 2013*; *Taetzsch et al., 2017*; *Tremblay et al., 2017*), implying that local, MU-independent, structural changes could occur. Finally, ALS is a non-cell autonomous disease where cell types other than motor neurons contribute to the disease pathogenesis (*Boillée et al., 2006a*; *Ilieva et al., 2009*). Contribution of various cell types known to interact with NMJs and motor axons has been suggested (*Arbour et al., 2015*; *Chiu et al., 2009*; *Keller et al., 2009*; *Lobsiger et al., 2009*; *Loeffler et al., 2016*; *Nardo et al., 2016*; *Turner et al., 2010*; *Wang et al., 2012*), which could contribute to NMJ loss. In this scenario, asynchronous branch-specific synaptic changes would take place over time within single MUs.

Altogether, these divergent paradigms highlight that the precise sequence of events taking place at the NMJ in ALS remains ill-defined. From a therapeutic viewpoint, this distinction is of high importance since it could pinpoint the NMJ as an accessible target that can be exploited within a time window between the onset of structural changes and the global degeneration of the MU, as previously suggested (*Cantor et al., 2018*; *Miyoshi et al., 2017*; *Pérez-García and Burden, 2012*). Here, we used longitudinal repeated in vivo imaging of single motor neuron axonal arbors and their NMJs to directly assess the time course and dynamics of structural changes during ALS progression. We

predict that a MU-dependent mechanism would result in the more or less synchronous degeneration of the whole MU over a brief period, while a contribution of local neuromuscular changes would result in the asynchronous loss of NMJs within a single MU.

## Results

### In vivo imaging of single MU arbors

SOD1$^{G37R}$ mice (lox$SOD1^{G37R}$) are well known for their slow disease progression, making it an ideal model to follow single MU changes over time (*Boillée et al., 2006b*; *Lobsiger et al., 2009*). To visualize and follow single MUs in control and ALS mice, we crossed SOD1$^{G37R}$ mice to animals expressing a cytoplasmic fluorescent marker in a small random subset of MNs (*Thy1*-YFP line H, *Figure 1A*) (*Feng et al., 2000*). Repeated in vivo imaging was performed on the *Tibialis anterior* muscle since numerous NMJs can be observed near the surface when exposed following a minimally invasive surgery (WT/YFP: N = 5; SOD1$^{G37R}$/YFP: N = 11). Postsynaptic sites were identified in vivo by staining nicotinic receptors (nAChR) with fluorescently labeled α-bungarotoxin (*Li et al., 2011*; *Turney et al., 2012*). Except for two cases (animal #23 and #21), all imaging sessions were performed at two–week intervals for up to 5–6 sessions (56–76 days). The same MU arbor (axon and its NMJs) was identified on every imaging session through the unique 'pretzel' shape of its NMJs, the branching pattern of the motor axon, and the position of nearby postsynaptic sites (*Figure 1A*, landmarks, Δ, *, •). NMJs were classified based on their innervation status over time (*Figure 1—figure supplement 1*).

We first confirmed that the same MU arbor and its NMJs were reliably imaged over several weeks without inducing any structural damage. We followed single MUs in WT/YFP animals and found that YFP-labeled axons stayed precisely overlapped with nAChR-rich postsynaptic sites over all imaging sessions performed (36/36 NMJs, 5/5 MU arbors, N = 5; *Figure 1C*). Importantly, very few changes in axonal branching pattern, postsynaptic nAChRs and presynaptic nerve terminal organization were observed. Consistent with previous reports (*Balice-Gordon and Lichtman, 1990*; *Li et al., 2011*; *Schaefer et al., 2005*; *Turney et al., 2012*), our results show that NMJs of WT mice remained innervated by the same MU and were very stable over long periods of time.

### Asynchronous NMJ dismantlement in SOD1$^{G37R}$ mice

We next performed imaging in symptomatic SOD1$^{G37R}$/YFP animals (all animals at or past disease onset) to investigate whether the pattern of NMJ denervation in ALS mice was synchronous or asynchronous. Of note, SOD1$^{G37R}$/YFP mice were phenotypically and histologically indistinguishable from SOD1$^{G37R}$ mice (*Figure 2—figure supplement 1*), showing that YFP expression did not alter the disease course and that YFP-expressing MNs are not more susceptible to the disease.

We found that YFP-labeled nerve terminals gradually retracted from some nAChR-rich postsynaptic sites (*Figure 2A–C,* for example, NMJ A1, A3, A4) while other axonal branches remained perfectly overlapped (e.g. NMJ A2). Some NMJs were even re-innervated by the MU (*Figure 2A,A4* inset, arrowhead, see below for more details) further suggesting that the MU arborization was not globally degenerating at this stage. Strikingly, most MUs in SOD1 mutant mice (16/19 arbors, 10/11 animals) behaved in a similar fashion, that is, losing NMJs over several imaging sessions rather than abruptly. These include partial (29/253 imaged NMJs, 12/19 arbors) and complete NMJ losses (38/253 imaged NMJs, 10/19 arbors). Thus, NMJ loss within a single MU starts as a slow asynchronous process, occurring independently of axonal degeneration.

### Local NMJ loss precedes global MU degeneration

A number of visualized motor axons and their remaining NMJs were eventually lost in a synchronous manner (7/19 MU arbors, 5/11 mice; *Figure 3A*) showing that global MU degeneration ultimately occurs. This phenomenon likely reflects an axonal or central event, such as MN death. Importantly, signs of NMJ denervation preceded most of these MU degenerations (*Figure 3B*, 6/7 MU arbors). In several cases (11/19 MU arbors), asynchronous NMJ dismantling started more than a month (≥ 3 sessions) prior to global degeneration of the MU, again showing that NMJ loss is initially a slow asynchronous process (*Figure 3B*). Hence, we identified two phases of NMJ loss within a MU: an initial slow local branch-specific dismantlement, followed by a sudden global axonal degeneration. These different patterns of MU degeneration are unlikely to be due to differences in MU types (*Frey et al.,*

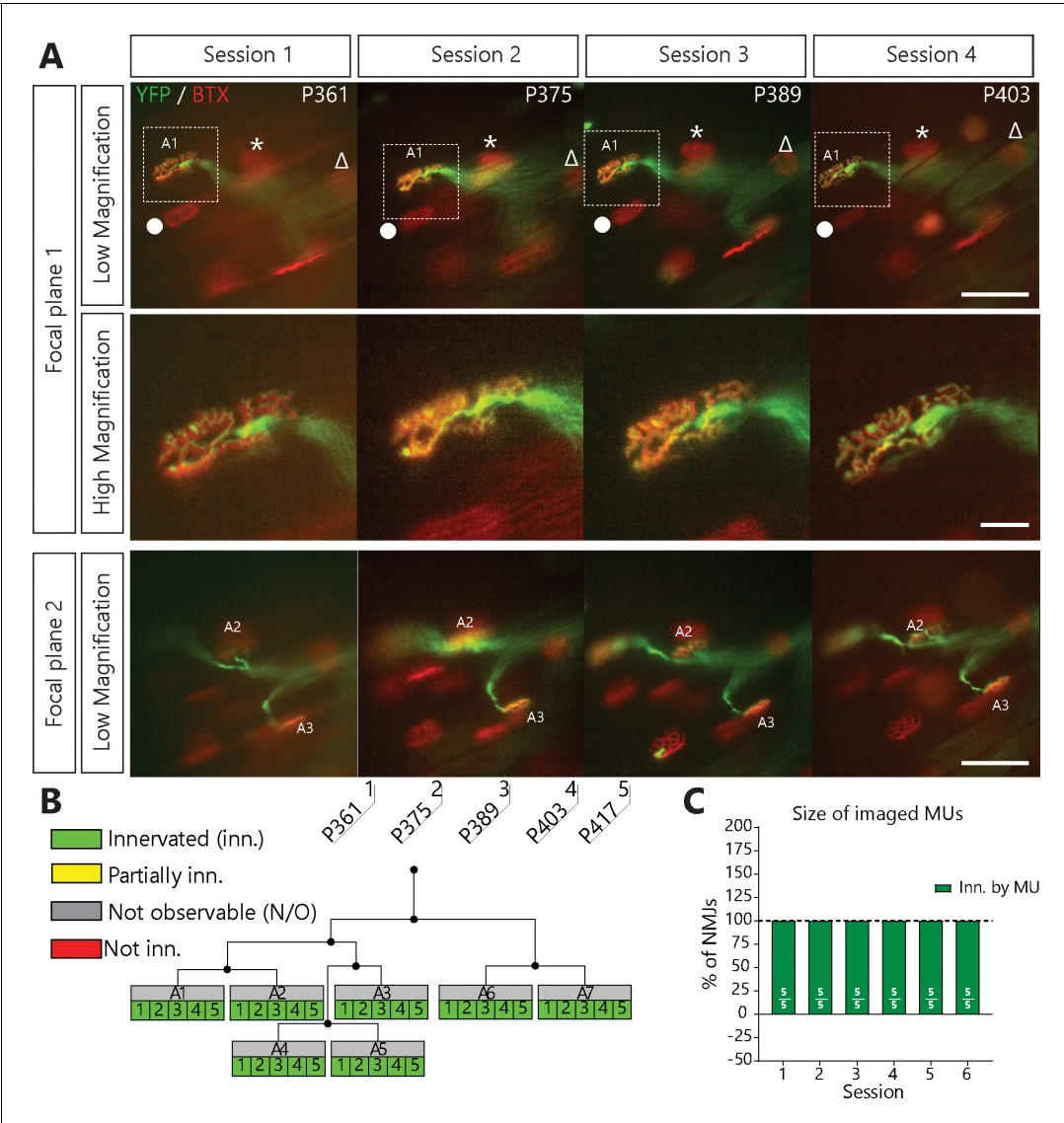

**Figure 1.** NMJs in a single motor-unit of WT/YFP mice are stable. (**A**) Images at different focal planes of the same MU arbor from a WT/YFP mice during four sessions, with a higher magnification on-focus image of one NMJ (digital zoom, dashed box in low magnification). Green: YFP-labeled axon; Red: nAChR. Symbols (Δ, *, •) identify landmarks used to confirm that the same region was imaged. (**B**) MU dynamic diagram of the MU shown in (**A**) showing that all NMJs within that motor arbor are stable in WT/YFP mice. Black lines and dots represent the axonal arborization and branching points respectively, while each box represents a single NMJ. (**C**) Histogram showing the average proportion of NMJs which are innervated (green) by the MU, showing that all MU preserved their NMJs in WT/YFP mice. Individual results for all MUs from WT mice (N = 5), including graphical representations, are included in *Figure 1—source data 1*. Details on how NMJs were classified, with representative examples, are presented in *Figure 1—figure supplement 1*. Scale bar, low magnification: 100 μm; high magnification: 25 μm.

DOI: https://doi.org/10.7554/eLife.41973.003

The following source data and figure supplement are available for figure 1:

**Source data 1.** Spreadsheet for the innervation status of all NMJs in each MU arbor imaged in WT/YFP mice and individual graphical representations.
DOI: https://doi.org/10.7554/eLife.41973.005
**Figure supplement 1.** Classification of NMJs within single MUs.
DOI: https://doi.org/10.7554/eLife.41973.004

---

*2000*; *Hegedus et al., 2007*; *Hegedus et al., 2008*; *Pun et al., 2006*). Indeed, fiber type (indicative of MU type) composition was examined on *Tibialis anterior* cross sections and revealed that nearly all superficial fibers in SOD1$^{G37R}$/YFP were fast-fatigable during the period of imaging (myosin heavy-chain isoforms type IIb or IIx; *Figure 3—figure supplement 1*).

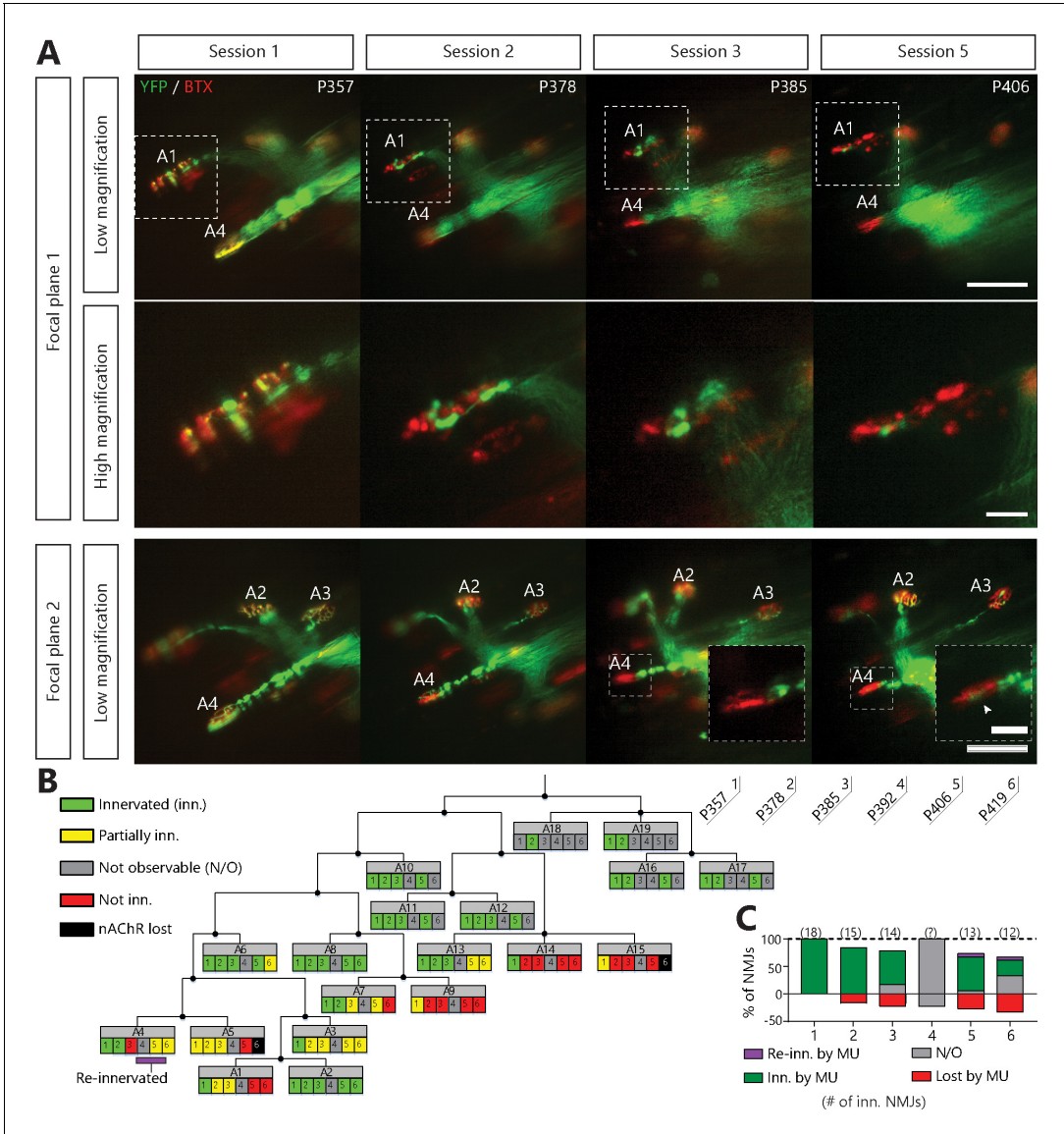

**Figure 2.** Single motor-unit degeneration is initially asynchronous and can last for several weeks in SOD1[G37R]/YFP mice. (**A**) Images at two different focal planes of the same MU arbor during four imaging sessions (1, 2, 3 and 5), with on-focus images at higher magnification of some NMJs (digital zoom, dashed boxes in low magnification). Green: YFP-labeled axon; Red: nAChR. Of note, A4 eventually gets partially re-innervated (inset, arrowhead, autologous reinnervation). (**B**) MU dynamic diagram showing that NMJs within this motor axon branch were lost asynchronously over imaging sessions. (**C**) Histogram showing the proportion of NMJs which are innervated (dark green), re-innervated (purple), not observable (gray) or lost (red) by the MU in A. (#): Number of NMJs observed in the MU arbor. Control experiments demonstrating that the SOD1[G37R]/YFP mice progress identically to SOD1[G37R] mice and that YFP expression does not exacerbate or affect motor neuron death are presented in *Figure 2—figure supplement 1*. Scale bar, low magnification: 100 μm; high magnification: 25 μm.

DOI: https://doi.org/10.7554/eLife.41973.006

The following source data and figure supplements are available for figure 2:

**Figure supplement 1.** Mating of floxSOD1[G37R] to thy1-YFP-H mice did not alter disease course or mutant SOD1 levels.

DOI: https://doi.org/10.7554/eLife.41973.007

**Figure supplement 1—source data 1.** Raw values of body weight, grip strength, α- and γ-motor neuron counts, percentages of YFP-expression motor neurons and NMJ innervation as depicted in *Figure 2—figure supplement 1B,C,E,F,H and I*, respectively.

DOI: https://doi.org/10.7554/eLife.41973.008

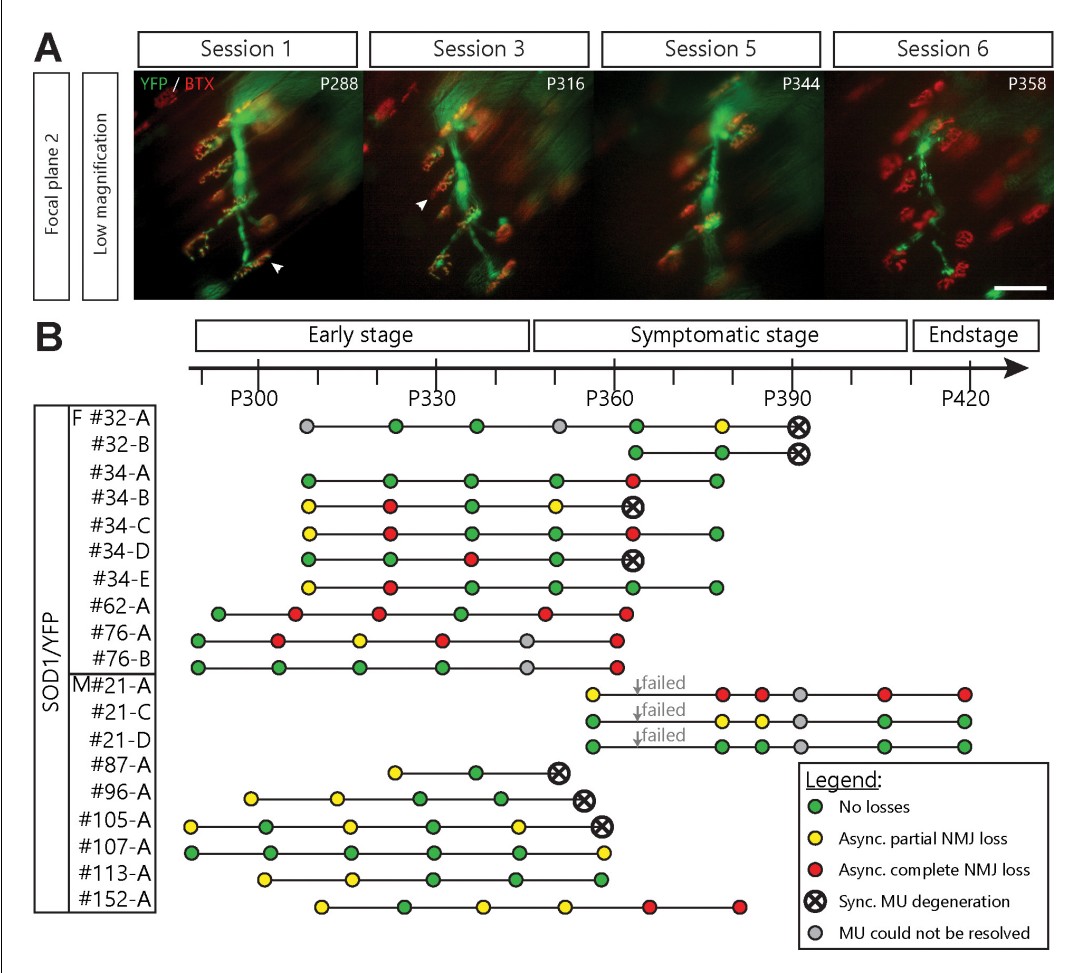

**Figure 3.** Asynchronous branch-specific dismantlement precedes synchronous motor-unit degeneration in SOD1[G37R]/YFP mice. (**A**) Example of another MU arbor showing two partially innervated NMJs (arrowheads) on session 1 and 2, before the whole motor axon degenerated on session 6. Note the presence of YFP fragments along the axonal tract. (**B**) Time course of repeated in vivo imaging in SOD1[G37R]/YFP mice showing sessions where no losses, at least one partial NMJ loss, at least one complete NMJ loss or synchronous MU degeneration were observed (green, yellow, red and black dots, respectively). Gray dots represent session where the MU could not be resolved while 'Failed' identifies session terminated for animal care reasons. Notice how synchronous global MU degenerations were almost always preceded by weeks of asynchronous branch-specific changes, even at later stages of the disease. The distribution of surface muscle fiber types (MU types) is presented in *Figure 3—figure supplement 1*, showing that these different patterns of degeneration are unlikely due to differences in MU types.

DOI: https://doi.org/10.7554/eLife.41973.009

The following figure supplement is available for figure 3:

**Figure supplement 1.** Motor-units are mainly fast-fatigable on the surface of the *Tibialis anterior* in SOD1[G37R]/YFP mice.

DOI: https://doi.org/10.7554/eLife.41973.010

## MUs expand their axonal arbor but do not reinnervate their lost NMJs

Surprisingly, a vast majority of MUs in SOD1[G37R]/YFP also sprouted toward nearby nAChR-rich post-synaptic sites, forming new synaptic contacts on muscle fibers that they did not innervate initially (MU expansions or 'heterologous re-innervation', *Figure 4A,B*; 76/253 NMJs, 14/19 arbors, 9/11 animals). These numerous expansions resulted in a net increase in MU size which masked the contemporaneous NMJ dismantlement (*Figure 4C*; 26.0 ± 10.2% loss vs 80.7 ± 32.2% expansions on session 6). Importantly, we found that over half of analyzed MUs both lost and formed new NMJs throughout the imaging process (*Figure 4—figure supplement 1*, 13/19 arbors, 8/11 animals), showing that these opposite processes can occur simultaneously within a single MU.

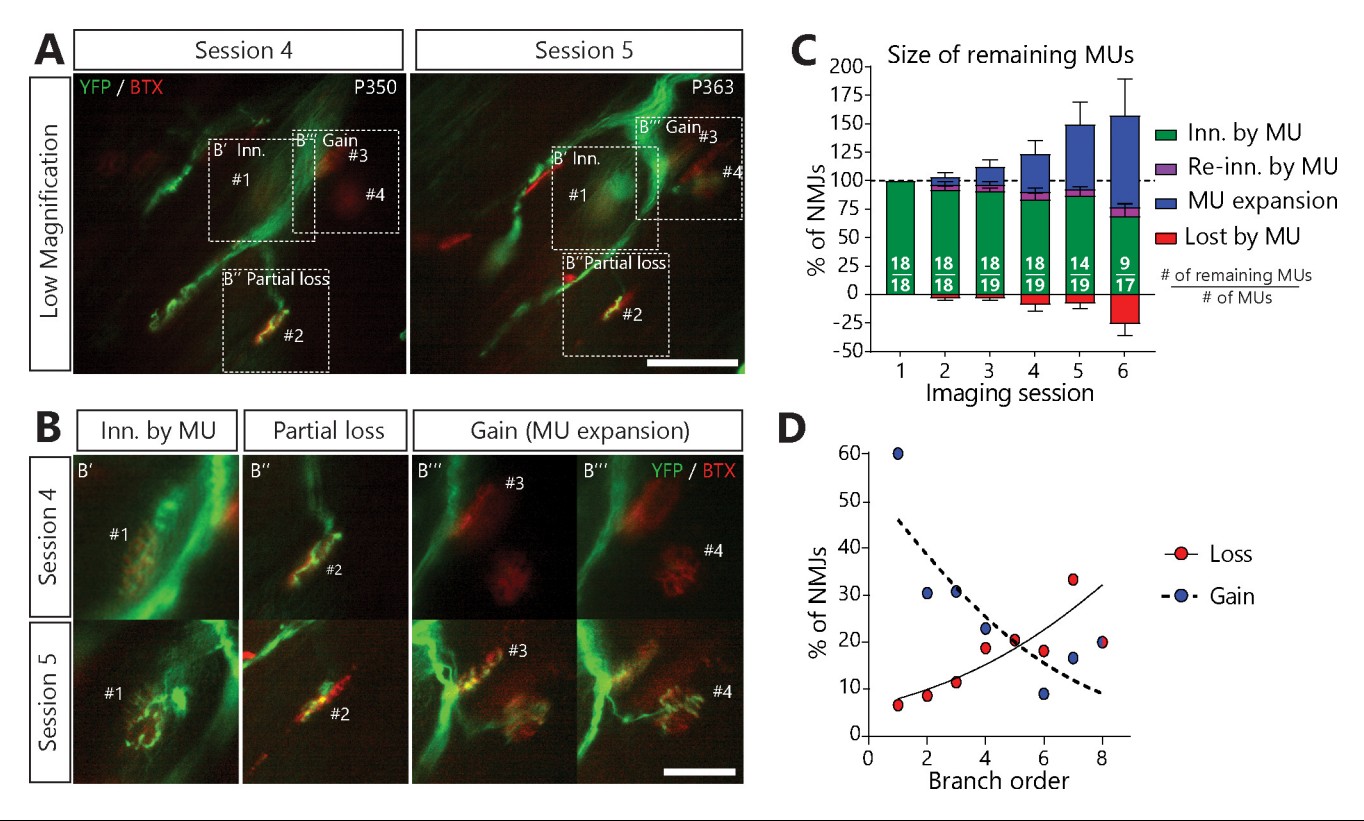

**Figure 4.** MUs retracted and expanded constantly during disease progression in SOD1$^{G37R}$/YFP mice. (**A**) (**B**) Images of a MU branch on two sessions (**A**) with on-focus high-magnification images of 4 NMJs (**B**) (dashed boxes in A), where one NMJ remained innervated (#1, B'), one was partially lost (#2, B'') and two others were newly innervated by this MU through axonal sprouting (#3 and #4, B'''; heterologous reinnervation) (Green: YFP-labeled axon; Red: nAChR). (**C**) Histogram showing the average proportion of NMJs from the initial pool which are innervated (green), re-innervated (purple, 'autologous reinnervation') or lost (red) by the MU and the proportion which are gained through expansions (blue, 'heterologous reinnervation'). Note that the overall size of MUs increased over time despite the loss of NMJs from their initial pool. Numbers in brackets represent the number of remaining MU arbors over the total number of MU arbors observed (N = 11). *Figure 4—figure supplement 1* illustrates how these opposing changes could alternate within the same MU over time. Controls showing that heterologous reinnervation (MU expansions) occurs on NMJs lost by the imaged MU are presented in *Figure 4—figure supplement 2*. (**D**) Correlation between MU expansions (blue) or asynchronous complete NMJ losses (red) and their branch order (n = 253; 19 arbors; N = 11) showing the inverse relationship between the propensity of axonal branches to expand (*logistic regression*, p=0.001) or to be lost (*logistic regression*, p=0.010). Each dot represents a data point while the solid and the dotted lines represent the logistic regression curves. Individual results for all MUs from SOD1 mice (N = 11), including graphical representations, are included in *Figure 4—source data 1*. The raw values for the graph in panel (**D**) are presented in *Figure 4—source data 2*. Data are presented as mean ± SEM. Scale bar, low magnification: 100 µm; high magnification: 25 µm.

DOI: https://doi.org/10.7554/eLife.41973.011

The following source data and figure supplements are available for figure 4:

**Source data 1.** Spreadsheet for the innervation status of all NMJs in each MU arbor imaged in SOD1/YFP mice and individual graphical representations.
DOI: https://doi.org/10.7554/eLife.41973.015

**Source data 2.** Spreadsheet including the number and the percentage of lost and gained NMJs as a function of their branch order.
DOI: https://doi.org/10.7554/eLife.41973.016

**Figure supplement 1.** Heat-map of all MU changes across all sessions.
DOI: https://doi.org/10.7554/eLife.41973.012

**Figure supplement 1—source data 1.** Percentage of change (gains, losses, total or delta) on each session relative to the first session.
DOI: https://doi.org/10.7554/eLife.41973.014

**Figure supplement 2.** Asynchronously lost postsynaptic sites do not become refractory to other MUs.
DOI: https://doi.org/10.7554/eLife.41973.013

Interestingly, however, there was an inverse relationship between the propensity of axonal branches to be lost or to expand (*Figure 4D*). Indeed, NMJs arising from distal axonal branches were more susceptible to being lost (*Figure 4D*, red dots, 38 events, n = 253, N = 11, *logistic regression, p*=0.010), while MU expansions were more likely to occur on proximal branches (mainly first-order branches) (*Figure 4D*, blue dots, 76 events, n = 253, N = 11, *logistic regression, p*=0.001). Hence, proximal branches preserve their capacity to re-innervate heterologous NMJs despite the ongoing degeneration of distal branches at the single MU level.

Regardless of this extensive compensation, few MUs re-innervated their own lost NMJs ('autologous re-innervation') and most partial losses were not repaired (*Figure 4C*, purple; 2/21 and 15/40 possibilities, respectively). Immunolabeling of all motor axons after the last in vivo session showed that some of these asynchronously lost NMJs had been reinnervated by other MUs (*Figure 4—figure supplement 2*, N = 4), confirming that these vacated postsynaptic sites were not refractory to reinnervation in general. This process led the initial NMJ pool of surviving MUs to gradually recede (*Figure 4C*, green) even though their total size remained stable or increased in most cases due to heterologous expansions (*Figure 4C*, blue).

Altogether, these results show that rather than being exclusively degenerating or compensating, MUs are highly dynamic during disease progression, with specific branches either expanding or disassembling.

## NMJ dismantlement seems to propagate

To gain further insight into the mechanisms underlying NMJ dismantlement, we next examined the spatial and temporal relationship between denervation events taking place within a MU. Indeed, local signals emanating from muscle fibers, glial cells or a damaged axonal branch (e.g. chemorepulsive molecules (*De Winter et al., 2006*; *Maimon et al., 2018*) or misfolded proteins (*Ayers et al., 2016*; *Grad et al., 2014*) could potentially propagate through the extracellular space and contribute to NMJ loss in ALS. If such local signals in the neuromuscular environment influence NMJ dismantlement, one would predict that these events may tend to propagate from one or several focal points. To test this possibility, we performed a Sholl-like analysis on all MU arbors bearing multiple complete NMJ losses (37 events out of 141 NMJs over 9 MU arbors, N = 6; see Materials and methods). Interestingly, asynchronous NMJ dismantling tended to propagate from the first lost NMJ (*Figure 5A and B*; *multivariate repeated-measures GLM with logistic distribution*, interaction between distance and time: p<0.001). Indeed, adjacent events (<300 μm away) occurred preferentially, but not exclusively, within the next two sessions ($T_2$), while remote events (>300 μm) occurred exclusively three or more sessions later ($T_3$+). These results unravel a propagation pattern and suggest that factors extrinsic to motor axons (i.e. the local neuromuscular environment) interact with the intrinsic properties of motor axons (e.g. susceptibility of specific branches), to shape the pattern of NMJ loss in ALS.

## Discussion

Our data reveal that single MNs initially lose NMJs asynchronously while they contemporaneously expand their axonal arbors to form new synaptic contacts in a mouse model of inherited ALS (*Figure 6*). The resulting increase in axonal arbor size masks the underlying NMJ dismantlement, effectively reconciling the 'dying-back hypothesis' with the observed MU enlargement in animal models and ALS patients (*Dantes and McComas, 1991*; *Dengler et al., 1990*; *Fischer et al., 2004*; *McComas et al., 1971*; *Pun et al., 2006*; *Schaefer et al., 2005*). Interestingly, these results are also in line with functional single MU recordings obtained from ALS patients identifying a majority of enlarged MUs and a subset of abnormally small MUs reflecting extensive NMJ dismantlement (*Dengler et al., 1990*; *Schmied et al., 1999*).

One of our key findings is that NMJ denervation in a MU is initially slow and takes place over more than a month in a single motor neuron in SOD1$^{G37R}$ mice. This time course reveals a long time-window after the onset of NMJ loss during which MNs are not globally degenerating and preserve their capacity to reinnervate NMJs. Peripheral stabilization of NMJs and MUs thus represents an attractive therapeutic target in the context of ALS (*Cantor et al., 2018*; *Miyoshi et al., 2017*), particularly when considering the accessibility of the neuromuscular system (i.e. no blood-brain-barrier). The propagating pattern of events and the asynchronous or multidirectional changes within a single

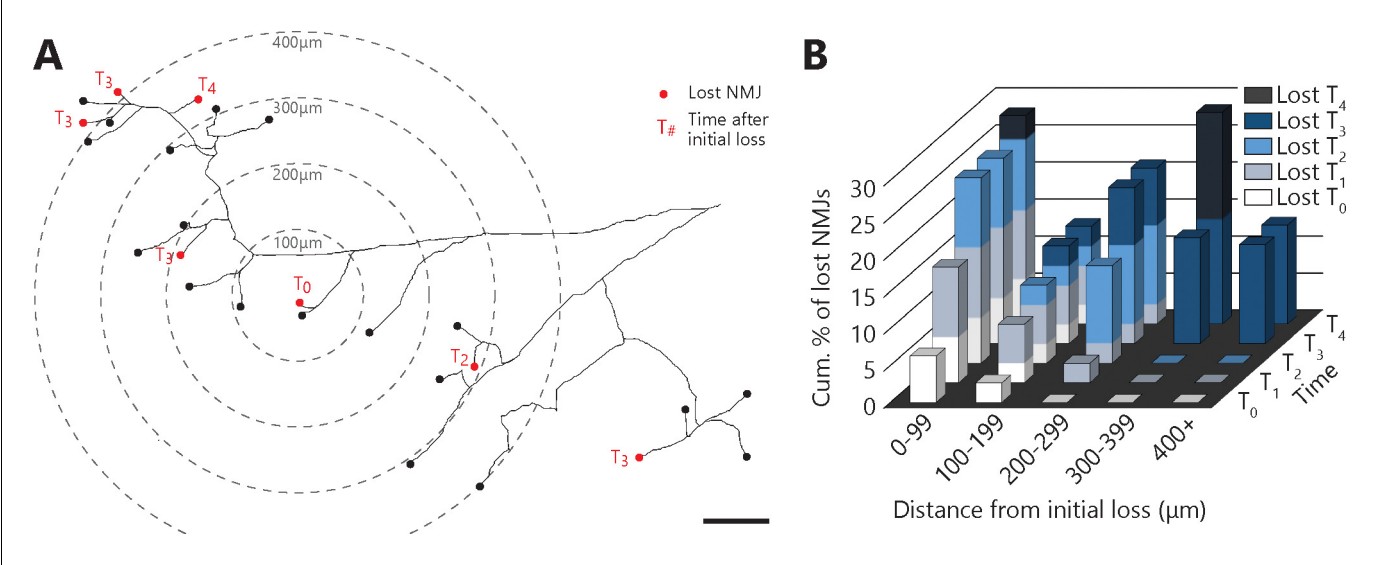

**Figure 5.** Denervation events tended to propagate from the first lost NMJ in the MU, with a higher susceptibility of distal branches. (**A**) Example of a full MU arbor tracing used to calculate the distance of every NMJ from the first one lost ($T_0$). Red dots indicate denervation events and their time (in session numbers) from the initial loss while black dots mark conserved NMJs. (**B**) Histogram showing the cumulative probability of an NMJ being lost as a function of distance and time from the initial loss, and for all MU arbors bearing multiple losses. Note how NMJs close to the initial loss (<300 µm) are frequently lost in the following sessions while distant NMJs are exclusively lost 3 or 4 sessions after (n = 141; 9 arbors, N = 6; *repeated measures GLM with logistic distribution*, effect of time: p=0.251; effect of distance: p=0.004; interaction between time and distance: p<0.001). The raw values for graphs in panel (**B**) are presented in *Figure 5—source data 1*. Scale bar: 100 µm.

DOI: https://doi.org/10.7554/eLife.41973.017

The following source data is available for figure 5:

**Source data 1.** Spreadsheet of the innervation status of all NMJs in each MU arbor with their distance from first lost NMJ in the arbor (in µm) and timing of subsequent losses relative to the initial losses ($T_0$, $T_1$, $T_2$, $T_3$ and $T_4$).

DOI: https://doi.org/10.7554/eLife.41973.018

MU suggest that neuromuscular factors extrinsic to motor axons (i.e. the local neuromuscular environment) affect or contribute to NMJ loss in ALS. These factors would presumably interact with the intrinsic vulnerability of axonal branches (distal vs proximal, etc.) to shape MU dynamism in ALS. Notably, chemo-repulsive molecules (*De Winter et al., 2006*; *Jokic et al., 2006*; *Maimon et al., 2018*; *Moloney et al., 2014*), availability of trophic factors (*Moloney et al., 2014*; *Taetzsch et al., 2017*), enhancement of NMJ stabilizing signals (*Miyoshi et al., 2017*; *Pérez-García and Burden, 2012*), secreted pathological proteins or dysfunctional glial cells (*Arbour et al., 2015*; *Arbour et al., 2017*) are mechanisms of therapeutic interest.

Another main finding is that MUs are not segregated as either degenerating or compensating (*Pun et al., 2006*; *Schaefer et al., 2005*). Rather, we observed a broad spectrum of MU phenotypes (*Figure 4—figure supplement 1*), with the majority continuously alternating between expansion and disassembly for weeks. This asynchronous branch-specific behavior contrasts with previous observations made by Schaefer and colleagues using single time point MU imaging in the fast-progressing SOD1[G93A] model (*Schaefer et al., 2005*). They reported that MUs with fragmented (degenerating) branches lacked thin axons (sprouts) while enlarged MUs lacked signs of degeneration. A number of observations can explain this apparent discrepancy. First, our time lapse analysis revealed that only some MUs underwent gains and losses in the same session (*Figure 4—figure supplement 1*), thus often giving the impression that they are either degenerating or compensating if only a single time point is considered. Second, we seldom observed a degenerating fragmented branch (9/38, *Figures 1 A,2A,A,4* for example), which led us to believe that the complete dismantlement of a single axonal branch can easily be missed if identified only by the fragmented axonal morphology. Another possibility would be that different MU types were considered in these studies, whereas the present study only focused on fast-fatigable MUs (MHC IIb and IIx fibers). However, *Pun et al., 2006*

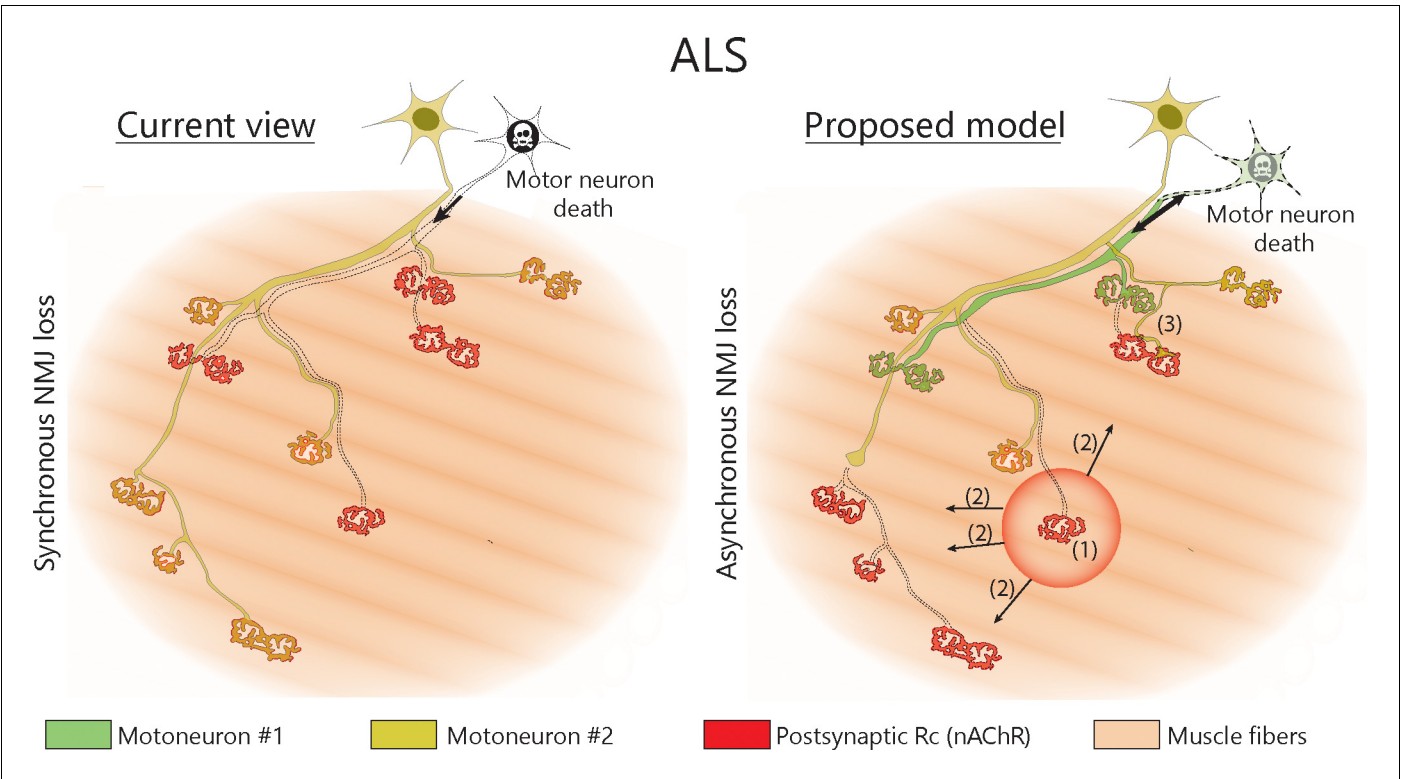

**Figure 6.** Proposed model of NMJ dynamism in ALS. NMJs within single MUs are lost asynchronously independently of motor axon degeneration (1). Local neuromuscular factors contribute to denervation events, which thus propagate from the first lost NMJ in the arbor (2), with distal branches being more susceptible than proximal ones. Nevertheless, MUs preserve their sprouting capability and extend toward heterologous, postsynaptic sites (3).
DOI: https://doi.org/10.7554/eLife.41973.019

reported no signs of compensatory reinnervation on the surface of the TA (same region as the present study) while *Schaefer et al., 2005* imaged a mixture of fast-fatigable and fast-resistant MUs (*Sternomastoid*, *Cleidomastoid* and *Clavotrapezius*), with the former presumably behaving similarly to what we describe here. Hence, we do not believe these discrepancies can be attributed to differences between MU types. Nevertheless, these studies raise the possibility that slower MU types may behave differently in ALS. One might speculate that the fast-resistant and slow MUs may be resistant to the asynchronous dismantlement and only exhibit late synchronous degeneration. Alternatively, these MUs may degenerate similarly to fast-fatigable MUs, but at a later time point, in part due to their small axonal arborization size (see below). Further examination of MU dynamism in a slow-twitch muscle, which could not be achieved here due to technical and ethical considerations, could shed light on these possibilities.

Interestingly, this pattern is reminiscent of post-natal synapse elimination at the NMJ, where the elimination of supernumerary axonal branches is also asynchronous and branch-specific within single MUs (*Keller-Peck et al., 2001*). This resemblance suggests that similar molecular mechanisms could be involved. A recent study has shown that selective axonal branch loss during post-natal synapse elimination was mediated by branch-specific microtubule destabilization (*Brill et al., 2016*). Interestingly, pharmacological stabilization of microtubules is beneficial to ALS mice (*Fanara et al., 2007*), suggesting that microtubule instability could underly the increased MU dynamism described here.

One unexpected finding was that MNs failed to re-innervate their lost NMJs, despite preserving their capacity to sprout and form new synaptic contacts onto other heterologous postsynaptic sites. Indeed, motor axons usually autologously reinnervate their original postsynaptic sites with great fidelity following an axonal lesion (*Nguyen et al., 2002*). It could be argued that MU expansions represent the autologous re-innervation of NMJs that were dismantled before session 1. However, this scenario seems unlikely considering that some YFP-negative axons reinnervated NMJs lost by the imaged YFP-positive motor axon (*Figure 4—figure supplement 2*). One possible explanation may

be that either the heterologous sprout itself or a molecular signal at the endplate prevents the original axon from reinnervating its target. Alternatively, asynchronous losses occur more frequently on distal branches, which seem intrinsically less likely to form sprouts (*Figure 4D*), thus reducing the likelihood of autologous reinnervation occurring.

Consistent with previous results (*Tallon et al., 2016*), we report that NMJs arising from distal branches in complex arbors (higher branch order) are more susceptible than proximal ones at the single MN level in ALS. These results support the notion that a larger axonal arborization size increases the energetic load on neurons, thus making them more vulnerable to pathological insults in neurodegenerative disease (*Le Masson et al., 2014*; *Pacelli et al., 2015*). One could speculate that the constant dynamism and the excessive MU expansions further increases the energetic load of MNs thus adding to their stress during disease progression.

## Concluding remarks

Overall, this detailed in vivo analysis reveals a previously unsuspected dynamism at the NMJ in an animal model of ALS. Our findings highlight a long temporal window between early branch-specific dismantling and global motor axon degeneration which could be therapeutically exploited in ALS (*Arbour et al., 2017*; *Moloney et al., 2014*).

# Materials and methods

**Key resources table**

| Reagent type (species) or resource | Designation | Source or reference | Identifiers | Additional information |
|---|---|---|---|---|
| Strain, strain background (*Mus muculus*, C57BL/6J) | lox*SOD1*$^{G37R}$ | PMID: 16741123; now also available from Jax mice (stock number 016149) | RRID:MGI: 3629226 | Originaly imported from Don W. Cleveland's facilities (UCSD). |
| Strain, strain background (*M. muculus*, C57BL/6J) | *Thy1*-YFP line H | Jax mice (stock number 003782) | RRID:MGI: 3497947 | |
| Antibody | Goat polyclonal anti-choline acetyl-transferase | EMD Millipore (AB144P) | RRID:AB_ 207951 | (1:100) |
| Antibody | Mouse monoclonal IgG1 anti-NeuN, clone A60 | EMD Millipore (MAB377) | RRID:AB_ 2298772 | (1:300) |
| Antibody | Rabbit polyclonal anti-S100β | Agilent Dako (Z0311) | RRID:AB_ 10013383 | (1:250) |
| Antibody | Chicken polyclonal anti-neurofilament M | Rockland Inc. (212-901-D84) | RRID:AB_ 11182576 | (1:2000) |
| Antibody | Mouse monoclonal IgG1 anti-synaptic vesicular protein 2 | DSHB (SV2) | RRID:AB_ 2315387 | (1:2000) |
| Antibody | Mouse monoclonal IgG2b anti-MHC1 | DSHB (BA-D5c) | RRID:AB_ 2235587 | (1:100) |

*Continued on next page*

*Continued*

| Reagent type (species) or resource | Designation | Source or reference | Identifiers | Additional information |
| --- | --- | --- | --- | --- |
| Antibody | Mouse monoclonal IgG1 anti-MHC2a | DSHB (SC-71c) | RRID:AB_2147165 | (1:200) |
| Antibody | Mouse monoclonal IgM anti-MHC2b | DSHB (BF-F3c) | RRID:AB_2266724 | (1:200) |
| Antibody | Mouse monoclonal IgM anti-MHC2x | DSHB (6H1s) | RRID:AB_1157897 | (1:10) |
| Antibody | Mouse monoclonal IgG1 anti-MHC all-but-IIx | DSHB (BF-35c) | RRID:AB_2274680 | (1:200) |
| Antibody | Goat polyclonal anti-mouse IgG1 DyLight 405 | Jackson Immuno Research (115-475-205) | RRID:AB_2338799 | (1:500) |
| Antibody | Donkey polyclonal anti-goat Alexa 594 | Jackson Immuno Research (705-585-147) | RRID:AB_2340433 | (1:500) |
| Antibody | Goat polyclonal anti-mouse IgG1 Alexa 647 | Jackson Immuno Research (115-605-205) | RRID:AB_2338916 | (1:500) |
| Antibody | Donkey polyclonal anti-chicken Alexa 647 | Jackson Immuno Research (703-605-155) | RRID:AB_2340379 | (1:500) |
| Antibody | Donkey polyclonal anti-rabbit Alexa 405 | Jackson Immuno Research (711-475-152) | RRID:AB_2340616 | (1:500) |
| Antibody | Goat polyclonal anti-mouse IgM Alexa 647 | Jackson Immuno Research (115-605-020) | RRID:AB_2338905 | (1:500) |
| Antibody | Goat polyclonal anti-mouse IgG2b Alexa 488 | Jackson Immuno Research (115-545-207) | RRID:AB_2338856 | (1:500) |
| Antibody | Goat polyclonal anti-mouse IgG1 Alexa 594 | Jackson Immuno Research (115-585-205) | RRID:AB_2338885 | (1:500) |
| Chemical compound, drug | Alexa 594-conjuguated α-Bungarotoxin | Thermofisher (B13423) | | (1:750) (1.33μg/mL) |

## Animals

WT/YFP ($SOD1^{-/-}$;$YFP^{+/-}$) and SOD1$^{G37R}$/YFP ($SOD1^{+/-}$;$YFP^{+/-}$) double transgenic mice were obtained by crossing transgenic males heterozygote for the human $SOD1^{G37R}$ transgene (flanked by LoxP sites; lox$SOD1^{G37R}$) to transgenic females heterozygote for the *Thy1*-YFP transgene (line H; [B6.Cg-Tg(Thy1-YFP)HJrs/J]; The Jackson laboratories, Bar Harbor, ME, stock number 003782) (*Feng et al., 2000*). Lox$SOD1^{G37R}$ mice have been described previously (*Boillée et al., 2006b*; *Lobsiger et al., 2009*). Single transgenic lines were maintained on a C57BL/6J background. Progenies were genotyped for the *SOD1* and the YFP transgenes by PCR performed on a tail sample or an ear punch extract. Mice of both sexes were used. Disease progression was monitored through weekly weighing and all-limb grip strength measurements (BioSeb, FL; BIO-GS3). Animals were sacrificed using a

lethal dose of isoflurane. All experiments were performed in accordance with the guidelines of the Canadian Council on Animal Care, the Comité de Déontologie sur l'Expérimentation Animale of Université de Montréal (protocol #18 – 040) and the CRCHUM Institutional Committee for the Protection of Animals (protocol #N16008CVV and #N15047ADPs).

## Repeated in vivo imaging

Procedures for repeated in vivo imaging of the *Tibialis anterior* muscle were adapted from previous reports (*Li et al., 2011*; *Schaefer et al., 2005*; *Turney et al., 2012*). Mice were anaesthetized with isoflurane (2 – 3% in 98 – 97% $O_2$) in an induction chamber and maintained under anesthesia using a breathing mask. Mice were placed on their side and a hind limb was immobilized on a custom-made stand. Eye drying was prevented by applying a lubricant (Vaseline). The *Tibialis anterior* muscle was exposed through an incision made on the external side of the leg (from ~5 – 10 mm proximal to the knee to ~5 mm proximal the ankle). The skin was drawn tight using four hooks (Guthrie Retractors; #17021 – 13; FST, Canada) mounted on custom-made blocks around the leg. The muscle was immersed in a physiological solution by regular irrigation with sterile lactated Ringer (B. Braun Medical, CDMV, Canada). Postsynaptic nAChRs were labeled with a non-blocking concentration of Alexa594-conjuguated α-Bungarotoxin (BTX, 5 μg/mL in sterile lactated Ringer for 10 min; Molecular Probes, Fischer Scientific, Canada). α-BTX was applied on the first session and was reapplied once (if necessary) only when the labeling was too dim, usually on session 5 or 6. Muscle contractions and synaptic transmission are known to be preserved at this concentration of α-BTX (*Arbour et al., 2015*; *Li et al., 2011*; *Turney et al., 2012*; *Zuo et al., 2004* and unpublished observations). NMJs near the surface were imaged using a 20X water immersion objective (0.4 NA, Nikon, Japan) mounted on an upright epifluorescent microscope (Nikon, Optiphot-2) equipped with a Neo-sCMOS camera (Andor, UK). Fluorescence excitation and emission were filtered using a Brightline Pinkel filter set optimized for CFP/YFP/HcRed (CFP/YFP/HcRed-3X-A-000; Semrock, NY). Images of the whole superficial MU axonal arborization at multiple focal planes were acquired using the Metafluor software (Molecular Devices, CA). After the imaging session, the wound was sutured using 5 – 0 or 6 – 0 vicryl suture (Johnson and Johnson), followed by tissue glue (GLUture; Abbot Laboratories, WPI, FL). Buprenorphine (3 μg/10 g body weight; Temgesic, CEVA Animal Health Ltd, UK) was administered by subcutaneous injections three times a day for 24 hr following the surgery. Antibiotics (0.1 mg/10 g body weight; Baytril; Bayer.Inc, Canada) were administered subcutaneously every 24 hr for 72 hr. Mice were given a small treat after each surgery (Nutella) to reduce stress and improve recovery. Except for two cases, imaging sessions were performed 2 weeks apart.

## In vivo images and data analysis

All available SOD1 animals with YFP-positive surface motor axons, were included in the in vivo analysis (16/44 mice tested). Only mice in which the MU arbor could be reliably followed for at least three sessions, with at least three NMJs near the surface, were included in the analysis (11/16 mice). NMJs which were too deep to be fully resolved were discarded. Images were analyzed using the Fiji software. First, focal planes from the same region were aligned using the 'StackReg' pluggin. MU branches were traced using the 'Simple Neurite Tracer' plugin. The clearest session (generally session 1) was chosen for tracing. All segments of the MU arbor (from each region imaged) were merged in Photoshop (Adobe) and BTX labelling (in the overlap between the regions) was used as a reference for alignment. Tracings were used as a map to identify each NMJ on subsequent imaging sessions. For the purpose of this analysis, two labeled branches that could not be undoubtedly linked to the same main motor axon were considered as independent MNs. Then, the state of innervation of each NMJ was determined and a MU dynamic diagram showing the state of each NMJ within the arborization on each session was created using Visio2013 (Microsoft). NMJs were then classified based on their dynamic changes (*Figure 1—figure supplement 1*). For figure representations, contrast was linearly adjusted to facilitate observations.

For the spatio-temporal analysis (Sholl-like analysis) of asynchronous NMJ dismantling (*Figure 5*), distance between NMJs and the initial loss was measured on the full MU tracings or, for a few cases, directly on the raw aligned images. The initial event (center) was defined as the first completely lost NMJ. When two or more NMJs were lost simultaneously, the one which first showed signs of denervation (partial loss) was defined as the center. If they could not be distinguished in this manner, the

analysis was carried out with multiple centers (2/9 MU arbors). For one animal (#34), several branches were fairly close to each other, that is in the same field of view (branches A and B, C and D). To more accurately assess the presence of a spreading pattern, a single center per field of view was determined. Branch order was determined using the motor-unit dynamic diagrams. MU expansions were assigned the same branch order as the branch from which they originated from, to determine likelihood of sprouting of axonal branches.

To create the heat map (*Figure 4—figure supplement 1*), the % of change was calculated by counting the number of gained or lost NMJs over the number of initial NMJs. Partial gains/losses counted as half an event. When an NMJ could not be observed, the % of change was spread across the sessions where it could have occurred.

## Myosin heavy chains labeling

Procedures for tissue preparation and immunostaining were done as previously described (*Tremblay et al., 2017*) Briefly, the *Tibialis anterior* was dissected in an oxygenated (95% $O_2$, 5% $CO_2$) Ringer REES solution and mounted in 10% Tragacanth (Sigma-Aldrich, Canada). Muscles were frozen in isopentane cooled to −80°C in liquid nitrogen. Transverse cryosections (10 μm) were incubated in blocking solution (10% normal donkey serum in PBS; Jackson Immunoresearch, PA) for 30 min, then with primary antibodies for 1 hr, and then with secondary antibodies for 1 hr. Sections were finally mounted in Prolong Gold antifade reagent (Molecular Probes, Fischer Scientific). Sections were rinsed three times with PBS 1X for 5 min between each step. Primary antibodies, all from the Developmental Studies Hybridoma Bank (DSHB, IA), were either mouse IgG1 α -MHC type IIa (SC-71c, 1:200), mouse IgG2b α-MHC type I (BA-D5c, 1:100) and mouse IgM α -MHC type IIb (BF-F3c, 1:200) or mouse IgM α -MHC type IIx (6H1s; 1:10) and mouse IgG1 α -MHC all but IIx (BF-35c, 1:200). Secondary antibodies were goat α-mouse IgG1 Alexa 594 (#115-585-205), goat α-mouse IgG2b Alexa 488 (#115-545-207) and goat α-mouse IgM Alexa 647(#115-605-020), all from Jackson Immunoresearch (all 1:500). Observations were made using an Olympus FV1000 or a Zeiss LSM880 confocal microscope with a 20X water immersion objective (N.A 0.95 or 1.0, respectively). Whole muscle sections were reconstructed by superimposing maximum intensity projections of each stack in Photoshop (Adobe).

## Whole-mount NMJ immunolabeling and motor neuron counts

Procedures for whole-mount muscle preparation, spinal cord sectioning and immunostaining were performed as previously described (*Tremblay et al., 2017*), with some minor adjustments. Mice were transcardially perfused with cold PBS 1X and 4% formaldehyde. Both *Tibialis anteriors* were then dissected and post-fixed for 20 min in 4% formaldehyde while the whole mouse was further fixed overnight at 4°C. Lumbar spinal cords were dissected, post-fixed for 2 hr, cryoprotected in a 30% sucrose-PBS solution for 72 hr at 4°C and frozen in cooled isopentane (−40 to −50°C).

Floating 30-μm-thick spinal cord cryosections were washed twice in PBS 1X, incubated in a donkey blocking solution (10% NDS, 0.3% Triton X-100 in PBS 1X) for 1 hr and then incubated overnight with primary antibodies against Choline Acetyl-Transferase (ChAT; 1:100; Goat; Millipore, Canada; AB144P) and Neuronal Nuclei (NeuN; 1:300; Mouse IgG1; Millipore; MAB377) in donkey blocking solution, then incubated with the secondary antibody donkey anti-goat Alexa 594 (1:500; #705-585-147; Jackson Immunoresearch) in donkey blocking solution for 1 hr, then incubated in a goat blocking solution (10% Normal Goat Serum, 0.3% Triton X-100 in PBS 1X) for 1 hr, then incubated with the goat anti-mouse IgG1 DyLight 405 secondary antibody (1:500; #115-475-205; Jackson Immunoresearch) in goat blocking solution for 1 hr and then finally mounted with Prolong Diamond antifade reagent (Molecular Probes). Sections were washed thrice with PBS 1X (5 min each) after each antibody incubation.

*Tibialis anterior* muscles were permeabilized in 100% cold methanol at −20°C for 6 min, then incubated for 1 hr in blocking solution (10% NDS, 1% of Triton-X100 diluted in PBS 1X). Muscles were then incubated with a rabbit anti-S100β antibody (1:250, Z0311, Dako) in blocking solution for 23 hr at 4°C, then with chicken anti-neurofilament M, (NF-M, 1:2000, Rockland Immunochemicals Inc) and mouse IgG1 anti-synaptic vesicular protein 2 (SV2, 1:2000, Developmental Studies Hybridoma Bank) for 23 hr at 4°C, then with the secondary antibodies goat anti-mouse IgG1 Alexa 647 (#115-605-205), donkey anti-chicken Alexa 647 (#703-605-155) and donkey anti-rabbit DyLight 405 (#711-

475-152) (all 1:500; Jackson Immunoresearch) simultaneously with Alexa594-conjugated-α-BTX (1.33 μg/ml, Molecular Probes, Fischer Scientific, Canada) for 2 hr. Finally, whole muscle preparations were mounted in Prolong Diamond antifade reagent (Molecular Probes). Muscles were washed six times (10 min each) in PBS 1X-Triton 0.01%.

Observations and image acquisition were performed on a Zeiss LSM 880 confocal microscope with a 20X water immersion objective (N.A. 1.0) or a 63X oil immersion objective (N.A. 1.4). No image manipulations were performed after acquisition, except for linear contrast adjustments for figure presentation. For motor neuron counts, MNs in both ventral horns were counted from 15 to 20 sections per animal, at least 90 μm apart. ChAT- and NeuN-positive cells in the ventral horn were counted as α-MNs while ChAT-positive and NeuN-negative cells were counted as γ-MNs as previously described (*Lalancette-Hebert et al., 2016*; *Tremblay et al., 2017*). The Allan Brain Atlas Mouse Spinal cord reference set was used as a reference to ensure that all analyzed sections were in the lumbar spinal cord. Results are expressed as the average number of cells counted per ventral horn in each animal.

## Statistical analysis

For the Sholl-like analysis, each lost NMJ represented an event in time (time after initial loss) and space (distance from the initial loss) occurring over a given number of trials (total number of NMJs). This type of data (i.e. number of observations over a number of trials as a % of events) follows a logistic distribution rather than a Gaussian distribution. Hence, a *multivariate repeated-measures Generalized linear models (GLM)* using a *logistic distribution* was created to test the effects of time and space on the likelihood of observing an event. The time after the initial loss was set as a 'within-NMJ' factor (the repeated measure) while distance from the initial loss was set as a continuous 'between-NMJs' variable. The raw measured distance was used rather than the range bins presented in *Figure 5*. For the branch order analysis, a similar approach was used. Each lost or gained NMJ represented an event occurring over a given number of trials (number of NMJs within that branch order) in eight conditions (branch orders). Hence, a *logistic regression* was used to test the effect of the branch order on the likelihood of an NMJ being loss (using the *univariate GLM with logistic distribution* command). The branch order was set as an ordinal variable (continuous). For these analyses, 'n' (the number of trials, i.e. the sample size) represent the number of NMJs observed while 'N' (the biological replicates) represents the number of animals. p-*Values* smaller than 0.05 (α = 5%) were considered statistically significant. Analyses were performed in the SPSS 24.0.0.0 (IBM) software.

When two independent groups were compared, *unpaired two-tailed t-tests* were performed using *Welch's correction* for unequal variance or a *Mann-Withney test* when data did not follow the assumption of a Gaussian distribution. Variances were compared using the F-test. When the effect of one variable was compared over time (motor behavior, *Figure 2—figure supplement 1*), *two-way ANOVAs* with repeated measures (*RM*) were used. For the post-test, Holm-Sidak's correction was used. For NMJ innervation (*Figure 2—figure supplement 1*), the effect of genotype was analyzed using a *GLM* with a logistic distribution and Holm-Sidak's correction was applied for the post-test as previously described (*Tremblay et al., 2017*). Importantly, a GLM with a logistic distribution was used for NMJ innervation instead of a *t-test* (as for motor neuron counts) because NMJ innervation data (# of denervated NMJs/ # of total NMJs for each animal) does not follow a Gaussian distribution. Unless otherwise stated, data are presented as mean ± SEM in the histograms and in the text. For these analyses, 'N' represents the number of biological replicates (animals, i.e. the sample size) while "n' represents the number of observations (number of NMJs unless otherwise stated). p-*Values* smaller than 0.05 (α = 5%) were considered statistically significant. These analyses were performed in the GraphPad Prism 7.0 software, with the exception of the analysis for the NMJ innervation was made in SPSS 24.0.0.0.

No sample size estimations were performed before the experiments. All available SOD1 animals which met our inclusion/exclusion criteria were included (see in vivo images and data analysis). For the NMJ innervation and motor neuron counts experiments, sample sizes similar to previous experiments were used (*Tremblay et al., 2017*). No test to detect outliers was performed.

## Acknowledgements

We thank Dr. Keith Murai and Dre. Janice Robertson for reading and commenting on the manuscript. We thank Sarah Peyrard and Joanne Vallée for essential help with animal husbandry, logistics for animal transfers between facilities and technical support. We also thank Dre. Danielle Arbour, Dr. Sébastien Barbat-Artigas and Alexandre St-Pierre-See for the invaluable discussions regarding data analysis, presentation and statistical testing. This work was funded by grants from the Canadian Institutes for Health Research (RR MOP-111070; ADP PJT-152934), Robert Packard Center for ALS Research (RR), Canadian Foundation of Innovation (RR, CVV), ALS Society of Canada (CVV), Muscular Dystrophy Association (CVV) and an infrastructure grant from the Fonds Recherche Québec-Santé to the GRSNC. CVV is an FRQS Senior Research Scholar. ÉM held a doctoral studentship from the ALS Society of Canada.

## Additional information

### Funding

| Funder | Grant reference number | Author |
|---|---|---|
| Canadian Institutes of Health Research | MOP-111070 | Richard Robitaille |
| Canadian Foundation for Innovation | | Christine Vande Velde Richard Robitaille |
| ALS Society of Canada | Doctoral Research Award | Éric Martineau |
| Muscular Dystrophy Association | | Christine Vande Velde |
| Fonds de Recherche du Québec - Santé | | Christine Vande Velde |
| Robert Packard Center for ALS Research, Johns Hopkins University | | Richard Robitaille |
| Canadian Institutes of Health Research | PJT-152934 | Adriana Di Polo |
| ALS Society of Canada | | Christine Vande Velde |

The funders had no role in study design, data collection and interpretation, or the decision to submit the work for publication.

### Author contributions

Éric Martineau, Conceptualization, Data curation, Formal analysis, Validation, Investigation, Visualization, Methodology, Writing—original draft, Writing—review and editing; Adriana Di Polo, Christine Vande Velde, Resources, Supervision, Funding acquisition, Methodology, Writing—review and editing; Richard Robitaille, Conceptualization, Resources, Supervision, Funding acquisition, Methodology, Writing—original draft, Project administration, Writing—review and editing

### Author ORCIDs

Éric Martineau http://orcid.org/0000-0002-5503-0798
Adriana Di Polo http://orcid.org/0000-0003-1430-0760
Richard Robitaille http://orcid.org/0000-0001-6628-0146

### Ethics

Animal experimentation: All experiments were performed in accordance with the guidelines of the Canadian Council on Animal Care, the Comité de Déontologie sur l'Expérimentation Animale of Université de Montréal (protocol #18-040) and the CRCHUM Institutional Committee for the Protection of Animals (protocol #N16008CVV and #N15047ADPs).

Decision letter and Author response
Decision letter https://doi.org/10.7554/eLife.41973.022
Author response https://doi.org/10.7554/eLife.41973.023

## Additional files

### Supplementary files
• Transparent reporting form
DOI: https://doi.org/10.7554/eLife.41973.021

### Data availability
All data generated or analysed during this study are included in the manuscript and supporting files. Source data has been provided for Figure 1, Figure 2 - Supplement 1, Figure 4, Figure 4 - Supplement 1, and Figure 5.

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
