## [Decision Letter]

[Editors’ note: a previous version of this study was rejected after peer review, but the authors submitted for reconsideration. The first decision letter after peer review is shown below.]

Thank you for submitting your work entitled "Dynamic neuromuscular remodeling precedes motor-unit loss in a mouse model of ALS" for consideration by *eLife*. Your article has been reviewed by three peer reviewers, one of whom is a member of our Board of Reviewing Editors, and the evaluation has been overseen by a Senior Editor. The reviewers have opted to remain anonymous.

Summary:

Our decision has been reached after extensive consultation among the reviewers. Based on these discussions and the individual reviews below, we regret to inform you that the consensus view that emerged is that we do not feel that the work as submitted is a good fit for *eLife*. We know that this is not happy news. That said, all the reviewers (and editors) agree that the images are remarkable. We would be willing to consider a new manuscript abbreviated to focus on what we see as the primary novelty of the your work: that denervation in a genetic form of ALS is not from a neuron/axon so damaged that it cannot regenerate, but rather that initial denervation is accompanied by simultaneous attempts at renervation.

If you would like to move forward with the manuscript in its present form, we wish you best of success at other journals. If you are willing to accept the proposal to streamline the effort as outlined above, *eLife* would be willing to consider a shortened version as a new submission.

*Reviewer #1:*

This manuscript from Robitaille and colleagues uses remarkable, sophisticated imaging to follow the same neuromuscular junctions for up to 76 days in mice in which denervation is driven by synthesis of an ALS-causing mutation in superoxide dismutase. The primary conclusions, which will be surprising to many in the research community and seem well known to the clinical community from the increase in muscle unit size in ALS patients, are that branch-specific degeneration is accompanied in the same axon by new sprouting. This insight is bundled with a very detailed, slowly paced text, that includes many aspects of lower interest (e.g., highlighting unexplained differences between males and females).

I started enthusiastic but could not sustain it. The images are remarkable but the text would have to be highly streamlined – to no more than a third of the current length – to be something that could be recommended to *eLife*.

*Reviewer #2:*

Muscle denervation is an early sign of disease in ALS and mouse models of ALS. Here, the authors repeatedly recorded images of single, labeled motor units to determine whether neuromuscular junctions from a single motor unit disassemble synchronously or asynchronously. As such, the authors sought to determine whether the motor nerve terminal or the motor neuron as a whole is the 'unit' of denervation. They report that synapses are disassembled asynchronously, indicating that the synapse rather than the neuron per se is the target for denervation.

Others (Lichtman/Sanes) raised a similar question in studying the mechanisms of synapse elimination during development and similarly found that the synapse behaves as the 'unit', as certain synapses from a single motor neuron are lost while other synapses from the same motor neuron are maintained (Keller-Peck et al., 2001). It would seem sensible to discuss the results reported here in light of these findings.

The reported findings are interesting and well-documented. I question however whether the findings are of broad interest and provide a sufficiently large advance in our knowledge and understanding of ALS to warrant publication in *eLife*.

1) The authors indicate that sub-saturating levels of fluorescent α-bungarotoxin were used to label and image nAchRs, but they provide no further information regarding how they determined that α-bungarotoxin binding was subsaturating and had no effect on the process of denervation.

2) The authors report that some denervated synapses were reinnervated by the same motor unit. They should supply this number/percent in the text.

3) The authors indicate that they imaged IIb and IIx myofibers, leaving open the possibility that motor nerve terminals on slower IIa muscle fibers behave differently. Did the authors examine other muscles in which surface IIa muscle fibers could be imaged to answer this question?

4) The authors indicate that the later loss of synapses from a single motor unit were more synchronous. It would help if the authors provided a graph/plot illustrating synchrony vs. asynchrony at different stages.

5) The authors write: 'Overall, these data reveal a propagation pattern which indicates that the local neuromuscular environment (Moloney et al., 2014) and the intrinsic susceptibility of axonal branches contribute to NMJ loss.' It would be helpful if the authors were more explicit about the meaning of 'local neuromuscular environment' and 'intrinsic susceptibility' and the implications of these conclusions.

6) The authors compare their findings to those of Pun et al. (2006). Did Pun et al. limit their analysis to IIb and IIx muscle fibers or might the differences between the studies be due to data in the Pun et al. study from IIa muscle fibers?

7) Since motor units were more likely to expand in female than male SOD1^G37R^ mice, and female mice showed a decline in grip strength earlier than male mice, do the authors believe – or have evidence – that the expansions are deleterious?

8) As the authors point-out, it is surprising 'that motor neurons failed to re-innervate their lost NMJs'. Indeed, one would expect motor axons, enveloped within a Schwann cell tube, would be constrained and directed back to original synaptic sites. Do the authors have data to indicate where the motor axons became 'lost'? Did the axons escape near the non-myelinating terminal Schwann cell or from myelinated tracks far from the synapse?

9) The authors write that 'Interestingly, sex-specific differences have been identified in other ALS mouse models and in patients, but they did not consistently point toward a faster or more severe disease in one sex'. This statement does not account for numerous studies showing that disease onset and death (more severe disease) occur earlier in male than female SOD1^G93A^ mice.

References:

1. The authors cite Nguyen et al. (2002) to indicate that axons reinnervate original synaptic sites, but more appropriate references would be studies form McMahan and colleagues (Rotschenker, Letinsky, Marshall, Sanes) in the 1970s.

2) The authors write: 'This observation strongly suggests that strategies aimed at stabilizing NMJs or enhancing autologous re-innervation could be of great therapeutic benefit in ALS.' Surprisingly, the authors don't cite two studies that describe such strategies: Miyoshi et al. (2017) and Cantor et al. (2018).

*Reviewer #3:*

The primary goal of this manuscript was to clarify whether neuromuscular junctions (NMJs) are dismantled in a synchronous or asynchronous manner prior to motor-unit deterioration in ALS. Using the well-characterized SOD1^G37R^ ALS mice expressing YFP in a subset of motor neurons, and through a thoroughly described and well-executed longitudinal imaging paradigm, the authors have firmly established that NMJs are lost in an asynchronous manner, and that this may proceed complete global motor unit degeneration by weeks. They note that while motor units continue to expand their axonal arbors over a long time period, they very infrequently re-innervate the NMJs that they have lost.

This paper is of great significance and adds greatly to our understanding of the temporal events occurring at NMJs prior to motor unit breakdown and motor neuron death in ALS. Of note is the in vivo rigorous analysis, which is not easy to do. The experiments are well designed and executed and the data nicely presented for the most part.

The many strengths of this work largely outweigh the few minor, and for the most part stylistic, weaknesses.

[Editors’ note: what now follows is the decision letter after the authors submitted for further consideration.]

Thank you for resubmitting your work entitled "Dynamic neuromuscular remodeling precedes motor-unit loss in a mouse model of ALS" for further consideration at *eLife*. Your revised article had been favorably evaluated by three peer reviewers, one of whom is a member of our Board of Reviewing Editors, and the evaluation has been overseen by Gary Westbrook as the Senior Editor.

Summary of reviewer comment:

I am happy to say that the reviewers and editors commend you on incorporating the changes that were requested, shortening it to focus on the most important findings related to denervation in ALS. All of us think that the work makes a novel contribution, with extraneous materials now removed, leaving a crisply paced text from which the novelty can be easily extracted by each reader.

---

## [Author Response]

[Editors’ note: the author responses to the first round of peer review follow.]

We would like to thank the reviewers and editors for their interest and constructive suggestions. We have performed an extensive revision of the manuscript according to their comments as indicated below in the detailed rebuttal. As a result, the manuscript is now about 50% shorter (2852 words down from 4947; three fewer main figures down to six, and two fewer figure supplements and 11 fewer references, new total of 52). This condensed version focuses on our finding that NMJ loss is asynchronous within single motor units and is accompanied by extensive compensatory sprouting, and the non-essential details have been removed. Furthermore, we have carefully revised the manuscript to address all other specific recommendations made by the reviewers. Please find below the point-by-point rebuttal to those comments.

Reviewer #1:This manuscript from Robitaille and colleagues uses remarkable, sophisticated imaging to follow the same neuromuscular junctions for up to 76 days in mice in which denervation is driven by synthesis of an ALS-causing mutation in superoxide dismutase. The primary conclusions, which will be surprising to many in the research community and seem well known to the clinical community from the increase in muscle unit size in ALS patients, are that branch-specific degeneration is accompanied in the same axon by new sprouting. This insight is bundled with a very detailed, slowly paced text, that includes many aspects of lower interest (e.g., highlighting unexplained differences between males and females).I started enthusiastic but could not sustain it. The images are remarkable but the text would have to be highly streamlined – to no more than a third of the current length – to be something that could be recommended to eLife.

We have significantly shortened the manuscript to focus on the main finding according to the reviewers’ and editors’ comments. This abbreviated version focuses almost entirely on our finding that NMJ loss is asynchronous within single motor units and is accompanied by extensive axonal sprouting. The more detailed analyses and controls have either been removed or their importance in the text was greatly reduced as to not divert the attention away from the main findings. We believe that the retained analyses and controls (Figure 2—figure supplement 1 and Figure 5) are of importance as they show respectively that YFP expression did not affect motor neuron degeneration (thus that observations made on these neurons are not artefactual) and that NMJ loss seems to propagate, hinting towards a contribution of factors extrinsic to motor axons.

Reviewer #2:[…] Others (Lichtman/Sanes) raised a similar question in studying the mechanisms of synapse elimination during development and similarly found that the synapse behaves as the 'unit', as certain synapses from a single motor neuron are lost while other synapses from the same motor neuron are maintained (Keller-Peck et al., 2001). It would seem sensible to discuss the results reported here in light of these findings.

We added a short paragraph in the Discussion (fourth paragraph) to mention this similarity and a potential mechanistic link between the two phenomena.

The reported findings are interesting and well-documented. I question however whether the findings are of broad interest and provide a sufficiently large advance in our knowledge and understanding of ALS to warrant publication in eLife.1) The authors indicate that sub-saturating levels of fluorescent α-bungarotoxin were used to label and image nAchRs, but they provide no further information regarding how they determined that α-bungarotoxin binding was subsaturating and had no effect on the process of denervation.

This concentration of α-bungarotoxin (5 µg/mL for 10 minutes) is routinely used in our laboratory to label postsynaptic receptors for a variety of in vivoand ex vivo physiological experiments, including electrophysiological recordings of synaptic activity. We have never observed blockade of muscle contractions in these conditions (Arbour et al., 2015), and synaptic transmission can still be recorded (unpublished data). This concentration was determined based on those used by Jeff Lichtman’s and Wesley Thompson’s groups for their in vivostudies (Li et al., 2011; Turney et al., 2012).

By comparison, we use concentrations over 100 µg/mL to completely block muscle contractions in mice (unpublished data). This value is similar to those used by other groups to obtain complete blockade in vivoin rats (Love and Thompson, 1999) and frogs (Wines and Letinsky, 1991), and incubations had to be repeated to maintain the blockage. Of note, in the present study, α-bungarotoxin was applied on the first session and was only reapplied once, only when the labeling was too dim, usually on session 5 or 6 (8 – 10 weeks after).

A statement was added in the Materials and methods section to clarify this aspect.

2) The authors report that some denervated synapses were reinnervated by the same motor unit. They should supply this number/percent in the text.

The fact that motor axons seldom reinnervate their lost NMJs (autologous reinnervation) is first mentioned in the text in the section on asynchronous NMJ dismantlement (“Asynchronous NMJ dismantlement in SOD1^G37R^ mice”) when describing the example provided in Figure 2A. No number/percent are provided here as the only purpose of that statement was to describe this particular MU and to illustrate that it was not globally degenerating. We elaborate more thoroughly on the frequency of these reinnervation events (number and percentages) in the section describing the contemporaneous sprouting of MUs (subsection “MUs expand their axonal arbor but do not reinnervate their lost NMJs”, Figure 4C, purple bars). The mention “see below for more details” was added after the first statement to clarify that this is an intended omission and that this result is fully presented in a more appropriate section.

3) The authors indicate that they imaged IIb and IIx myofibers, leaving open the possibility that motor nerve terminals on slower IIa muscle fibers behave differently. Did the authors examine other muscles in which surface IIa muscle fibers could be imaged to answer this question?

Unfortunately, to our knowledge. no hindlimb muscle with surface IIa and I fibers in mice is suitable for the repeated in vivo imaging procedure. The *Soleus* muscle has a majority of surface type I and IIa fibers in mice and can be imaged in vivo, but the procedure to access it is very invasive and involves lifting the muscle with a spatula (Kang et al., 2014). This procedure can potentially induce muscle damage if repeated at two-week interval, which could affect the results. Furthermore, the surgery-induced fibrosis would likely limit our ability to accurately resolve NMJs on later sessions. This was already a limiting factor on some sessions when using a less invasive procedure (Figure 3B – “Not observable”). Alternatively, the *Sternomastoid*, a neck muscle, has some surface IIa fibers and has been frequently used by Sanes and Lichtman’s groups. However, the surgery is also very invasive, also involves lifting the muscle and requires the mice to be intubated. This intubation would not be well tolerated by aged symptomatic ALS mice if repeated every two weeks.

4) The authors indicate that the later loss of synapses from a single motor unit were more synchronous. It would help if the authors provided a graph/plot illustrating synchrony vs. asynchrony at different stages.

We thank the reviewer for this suggestion. We added the timeline presented in Figure 3, showing when asynchronous and synchronous losses were observed in each MU as a function of the age of the animal. This plot illustrates that asynchronous NMJ loss starts long before synchronous MU degeneration at the single MU level. However, asynchronous losses continued to be observed in surviving MUs, even at later stages.

5) The authors write: 'Overall, these data reveal a propagation pattern which indicates that the local neuromuscular environment (Moloney et al., 2014) and the intrinsic susceptibility of axonal branches contribute to NMJ loss.' It would be helpful if the authors were more explicit about the meaning of 'local neuromuscular environment' and 'intrinsic susceptibility' and the implications of these conclusions.

We clarified the statement in the subsection “NMJ dismantlement seems to propagate”. The second paragraph of the discussion also now elaborates more clearly on this idea.

6) The authors compare their findings to those of Pun et al. (2006). Did Pun et al. limit their analysis to IIb and IIx muscle fibers or might the differences between the studies be due to data in the Pun et al. study from IIa muscle fibers?

We are unfortunately unsure which reference to Pun et al. (2006) the reviewer is referring to. We believe he is possibly alluding to the subsection “Local NMJ loss precedes global MU degeneration”. We do not believe differences between the studies are due to type IIa being considered in Pun et al. since certain muscle regions they studied only contained IIb/IIx fibers (Class 1 sub compartment), notably the surface of the TA. Their study showed that signs of compensatory sprouting were restricted to regions containing S (I fibers) and to some extent FR MUs (IIa fibers). However, we show here that compensatory sprouting and asynchronous NMJ loss occur concomitantly with a single MU at the surface of the TA, a region containing mostly FF MUs.

We are unsure as to why our results differ from those of Pun et al. (2006). One possible explanation for this difference is that they used SOD1^G93A^ mice which progress much faster than SOD1^G37R^ mice. Therefore, the time window to observe these changes may be narrower and their study may not have had the temporal resolution to observe these events. This difference may also arise from differences in the methodology used to evaluate compensatory sprouting: our repeated in vivoimaging vs. their immunohistochemistry at various timepoints.

7) Since motor units were more likely to expand in female than male SOD1^G37R^ mice, and female mice showed a decline in grip strength earlier than male mice, do the authors believe – or have evidence – that the expansions are deleterious?

This is not relevant to the current version of the manuscript.

8) As the authors point-out, it is surprising 'that motor neurons failed to re-innervate their lost NMJs'. Indeed, one would expect motor axons, enveloped within a Schwann cell tube, would be constrained and directed back to original synaptic sites. Do the authors have data to indicate where the motor axons became 'lost'? Did the axons escape near the non-myelinating terminal Schwann cell or from myelinated tracks far from the synapse?

This is a very interesting point. When the axon retracted, it seemed to retract back to the main axonal branch/tack. When MU expansions (heterologous reinnervation) occurred, motor axons mostly formed axonal sprouts from nearby axonal tracks/branches (presumably myelinated). Terminal sprouts through the terminal non-myelinating Schwann cells were rare events in our observations (8/75 expansions). Based on the images in Figure 3—figure supplement 2A2/B2 and A4/B4, it would seem that the new heterologous axon uses the old Schwann cell tube as it clearly follows a similar path. This behavior was also described by Nguyen et al. (2002 – Figure 4) in the rare occasions that the incorrect axon (heterologous) would reinnervate an NMJ after a nerve lesion. What was surprising to us was how frequently this occurred in ALS mice compared to autologous reinnervation. Possible explanations for this phenomenon are addressed the Discussion.

9) The authors write that 'Interestingly, sex-specific differences have been identified in other ALS mouse models and in patients, but they did not consistently point toward a faster or more severe disease in one sex'. This statement does not account for numerous studies showing that disease onset and death (more severe disease) occur earlier in male than female SOD1^G93A^ mice.

This is not relevant to the current version of the manuscript.

References:1. The authors cite Nguyen et al. (2002) to indicate that axons reinnervate original synaptic sites, but more appropriate references would be studies form McMahan and colleagues (Rotschenker, Letinsky, Marshall, Sanes) in the 1970s.

The studies by McMahan and colleagues showed that the localisation of synaptic sites on muscle fibers was “marked” by the basal lamina after denervation. They showed that motor axons reinnervated these synaptic sites (“original sites”) as opposed to forming entirely new synaptic contacts at other locations on the muscle fibers. However, these studies did not show that a single motor axon would reinnervate the same set of NMJs that originally belonged to its motor unit. This was first shown by Nguyen et al. (2002) using the *thy1*-YFP-H mouse (90% fidelity).

2) The authors write: 'This observation strongly suggests that strategies aimed at stabilizing NMJs or enhancing autologous re-innervation could be of great therapeutic benefit in ALS.' Surprisingly, the authors don't cite two studies that describe such strategies: Miyoshi et al. (2017) and Cantor et al. (2018).

Thanks for the suggestions. We added these references to the relevant section in the Discussion.